# Cognitive Reserve, Early Cognitive Screening, and Relationship to Long-Term Outcome after Severe Traumatic Brain Injury

**DOI:** 10.3390/jcm11072046

**Published:** 2022-04-06

**Authors:** Natascha Ekdahl, Alison K. Godbolt, Catharina Nygren Deboussard, Marianne Lannsjö, Britt-Marie Stålnacke, Maud Stenberg, Trandur Ulfarsson, Marika C. Möller

**Affiliations:** 1Centre for Research and Development, Uppsala University/County Council of Gävleborg, 801 88 Gävle, Sweden; marianne.lannsjo@regiongavleborg.se; 2Department of Clinical Sciences, Karolinska Institutet, 182 88 Stockholm, Sweden; alison.godbolt@regionstockholm.se (A.K.G.); catharina.nygren-deboussard@regionstockholm.se (C.N.D.); marika.moller@ki.se (M.C.M.); 3Department of Rehabilitation Medicine, Danderyd University Hospital, 182 88 Stockholm, Sweden; 4Department of Neuroscience, Rehabilitation Medicine, Uppsala University, 751 24 Uppsala, Sweden; 5Department of Community Medicine and Rehabilitation, Umeå University, 901 85 Umeå, Sweden; brittmarie.stalnacke@rehabmed.umu.se (B.-M.S.); maud.stenberg@rehabmed.umu.se (M.S.); 6Department of Rehabilitation Medicine, Sahlgrenska University Hospital, 405 30 Gothenburg, Sweden; trandur.ulfarsson@vgregion.se

**Keywords:** traumatic brain injury, cognition, neuropsychology, patient outcome assessment, executive function, education, prognosis

## Abstract

The objective was to investigate the relationship between early global cognitive functioning using the Barrow Neurological Institute Screen for Higher Cerebral Functions (BNIS) and cognitive flexibility (Trail Making Test (TMT), TMT B-A), with long-term outcome assessed by the Mayo-Portland Adaptability Index (MPAI-4) in severe traumatic brain injury (sTBI) controlling for the influence of cognitive reserve, age, and injury severity. Of 114 patients aged 18–65 with acute Glasgow Coma Scale 3–8, 41 patients were able to complete (BNIS) at 3 months after injury and MPAI-4 5–8 years after injury. Of these, 33 patients also completed TMT at 3 months. Global cognition and cognitive flexibility correlated significantly with long-term outcome measured with MPAI-4 total score (r_BNIS_ = 0.315; r_TMT_ = 0.355). Global cognition correlated significantly with the participation subscale (r = 0.388), while cognitive flexibility correlated with the adjustment (r = 0.364) and ability (r = 0.364) subscales. Adjusting for cognitive reserve and acute injury severity did not alter these relationships. The effect size for education on BNIS and TMT scores was large (d ≈ 0.85). Early screenings with BNIS and TMT are related to long-term outcome after sTBI and seem to measure complementary aspects of outcome. As early as 3 months after sTBI, educational level influences the scores on neuropsychological screening instruments.

## 1. Introduction

Traumatic brain injury (TBI) is a major cause of lifelong disability in young adults [1,2]. The most severe form of TBI (sTBI) is characterized by great variance in outcome, from death to favorable outcome [3]. Predictors of long-term outcome for individual patients is uncertain, but long-term follow-up studies suggest that a combination of demographic, injury-related, and cognitive factors contribute [4,5,6,7]. Cognitive deficits are common following sTBI and affect work, leisure, and daily living activities [8]. Measuring cognitive deficits using a full neuropsychological assessment is time-consuming, and in the early stages after sTBI, it might not be feasible due to physical injuries and patients’ lack of stamina. A shorter screening of cognitive functions might therefore be preferable, but it needs to be ensured that shorter screenings capture cognitive functions important for outcome.

One easily administered cognitive screening instrument is the Barrow Neurological Institute Screen for Higher Cerebral Functions (BNIS [9]). Previous studies have shown that BNIS is related to outcome after sTBI measured with the Glasgow Outcome Scale-Extended (GOSE) [7,10]. Since TBI may affect many areas of functioning as well as community integration and emotional adjustment, it would be of value to measure outcome with more detailed outcome scales. The Mayo-Portland Adaptability Index (MPAI-4 [11]) has been developed specifically to measure these aspects of outcome after brain injury and has been demonstrated to be a valid and reliable instrument [12,13]. To our knowledge, no studies have been published relating BNIS to more detailed measures of long-term outcome after sTBI. Additionally, BNIS does not include any specific measures of executive functions. Given that deficits in executive functions are related to functional outcome after TBI, a brief executive test, such as Trail Making Test (TMT [12]), could be used in order to complement BNIS [14,15].

Age and acute injury severity consistently play a part in outcomes after sTBI [16]. Additionally, cognitive reserve, usually approximated by educational level, has been found to influence both the score on neuropsychological tests, including BNIS, and the outcome after sTBI [10,12,16,17]. However, a relationship between cognitive reserve and a test score does not automatically imply a relationship between test score and recovery. When investigating the relationship between cognitive screening and functional outcome after sTBI, it is therefore important to take age, injury severity, and cognitive reserve into consideration. Following sTBI even though most improvement is believed to take place within the first year, changes in functional outcome, both improvements and deterioration, can continue for several years. Given that TBI often affects young individuals, who are expected to live for decades with their injury, it is important to conduct studies with a longer follow-up interval [18]. In the Swedish health care system, 3 months after injury most patients are still undergoing inpatient rehabilitation. It is at this time point that discussion about likely long-term outcomes often becomes relevant for patients and relatives as they start to plan for life after hospital care. A better understanding of factors contributing to long-term outcomes would be of use to patients, relatives, and health-care staff in planning for continued rehabilitation and support services.

In the present study, the primary aim was to investigate the relationship between findings from early cognitive screening, using BNIS and TMT, and long-term (5–8 year) outcome assessed with MPAI-4 in sTBI. A secondary aim was to investigate whether cognitive reserve, as approximated by educational level, age, and acute injury severity, influences this relationship.

## 2. Materials and Methods

This study is a 5–8-year prospective longitudinal observational study of patients included in the multicenter research project “Probrain”. Probrain recruited patients (*n* = 114) during initial neurosurgical care from five neurosurgical intensive care units in Sweden and one in Iceland from January 2010 to December 2011. Follow-up was performed at the Swedish units from September 2016 to September 2018. Inclusion criteria for the Probrain study were as follows:Severe nonpenetrating, TBI, with a lowest nonsedated Glasgow Coma Scale (GCS) [19] score of 3–8 in the first 24 h after injury;Age at injury: 18–65 years;Injury requiring neurosurgical intensive care or collaborative care with a neurosurgeon in another intensive care unit.

Exclusion criterion was death within 3 weeks of injury. For detailed methodological information, please see earlier published studies [10].

Inclusion for patients in this study was a completed BNIS assessment at 3 months after injury and a completed MPAI-4 assessment at follow-up 5–8 years after injury.

### 2.1. Participants

In the long-term follow-up, 63 patients from the five neurotrauma centers from Sweden participated. Due to logistical reasons, two centers followed up a randomized sample of patients, resulting in seven patients not being invited to participate in the 5–8-year follow-up. MPAI-4 was completed by 54 patients, of whom 41 had completed BNIS 3 months after injury. For more details, see Figure 1.

There were no significant differences in age, gender, or educational level between the 41 patients included in the current study and the 73 excluded patients. However, the included patients had significantly higher acute GCS scores compared with those excluded.

### 2.2. Procedure

Patients were evaluated at 3 weeks, 3 months, 1 year, and 5–8 years after injury (mean of 6.6 years). In the current study, data from the 3-month screening and the 5- to 8-year follow-up were used. The 3-month time point was chosen as an earlier time point would result in too few patients being able to complete screening, and a later time point would be less relevant from a clinical standpoint, as patients are more likely to be discharged from the hospital at 1 year after injury. Furthermore, a previous article from the Probrain project found that cognition was rather stable between the 3-month follow-up and the 1-year follow-up, thus making it redundant to use data from both time points [10]. Assessments took place in the patient’s current care setting or in a local rehabilitation outpatient department. The 5- to 8-year follow-ups were conducted either in combination with a visit to the local outpatient department or by post. Data regarding education were obtained by interviews with patients and/or significant others. Educational level was dichotomized as high (≥12 years) and low (<12 years). At the 3-month screening, patients were interviewed, and the BNIS was administered, by either a clinical neuropsychologist or a physician specializing in rehabilitation medicine. GOSE was assessed, and MPAI-4 was administered at the 5- to 8-year follow-up. In case MPAI-4 was administered both in connection with a visit to the care facilities and by post, the highest score was chosen.

### 2.3. Instruments

#### 2.3.1. BNIS

The BNIS is a cognitive screening test of global cognitive functioning, encompassing speech and language function, orientation, attention/concentration, visuospatial and visual problem solving, memory, affect, and awareness [9]. The test includes a prescreen test with a maximum score of 9, and patients must achieve at least two points on each of the items for the assessment to continue. A full BNIS has a maximum score of 50, where higher scores reflect a higher level of functioning. If the total BNIS score is below 47, further cognitive investigation is recommended. BNIS takes approximately 15–20 min to complete.

#### 2.3.2. TMT

TMT assesses attention, processing speed, sequencing, mental flexibility, and visual-motor skills [12]. TMT contains two parts: In part A, numbers are required to be connected as fast as possible in numerical order. Part B is similar, but now the subject is required to alternate between numbers and letters, adding a component of executive functioning. The primary outcome variable is completion time. By calculating the difference between the completion time of TMT B and TMT A, a measure of cognitive flexibility is produced [20]. TMT has been proven to be sensitive to neurological impairment [12].

#### 2.3.3. MPAI-4

MPAI-4 consists of 30 questions aimed at assessing commonly occurring difficulties after brain injury. The score ranges from 0 to 111, where a lower score indicates better recovery. The instrument consists of three subscales: ability index, adjustment index, and participation index. The ability index measures sensory, motor, and cognitive abilities; the adjustment index measures mood and interpersonal interaction; and the participation index measures social contacts, initiation, work/school, and money management. Previous studies have demonstrated good person reliability (0.92) and item reliability (0.94) [11]. MPAI-4 is linked to the International Classification of Functioning, Disability, and Health and is an established tool for investigating long-term functional outcome after TBI [21]. The instrument covers areas of physical, cognitive, emotional, behavioral, and social problems that persons with brain injury can encounter and furthermore contains assessments of areas where problems commonly arise when patients are reintegrated in society [11].

#### 2.3.4. CRASH Model

In order to control for acute injury severity, the acute injury composite (corticosteroid randomization after significant head injury, CRASH) was used, representing a % risk of unfavorable outcome at 6 months, as calculated and used in previously published studies on the Probrain material [22]. The crash composite score includes data collected within the first 24 h regarding GCS, pupillary reaction, presence of major extracranial injury, age, country, and five CT-brain features [23]. A higher score indicates a greater risk of unfavorable outcome.

#### 2.3.5. GOSE

GOSE score spans from 1 (dead) to 8 (upper good recovery). Traditionally, a score in the range of 1–4 is considered an unfavorable outcome, and a score in the range of 5–8 is considered a favorable outcome. The GOSE has good interrater reliability and validity and is an established measure of global outcome after traumatic brain injury [24]. GOSE was included as a descriptive variable in order to more fully depict the patient group.

### 2.4. Analysis

Statistical analyses were computed using the Jamovi statistical software [25]. A significance level of 0.05 was used for all statistical tests. Differences in demographic characteristics between groups were analyzed with Student’s *t*-test for parametric data and a Mann–Whitney and χ2 test for nonparametric data. For measures of effect size, Cohen’s *d* was used, where 0.2 is considered a small effect size, 0.5 medium, and 0.8 large. In order to analyze the relationship between results on neuropsychological screening and outcome according to MPAI-4, correlation analysis with Spearman’s rho was used. To investigate the effect of the controlling variables on the relationship between results on neuropsychological tests and outcome, linear regression was used. Since the MPAI-4 subscales are not independent of each other, linear regression on the total score was not used. Three linear regressions were computed for BNIS and cognitive flexibility separately, one for each MPAI-4 subscale score, which was used as the dependent variable, in total six models. CRASH and educational level were added as independent variables to adjust for age, injury severity, and cognitive reserve. Age is included in the CRASH model and, therefore, not entered separately in the linear regression model.

### 2.5. Ethics

The study was approved by the Regional Ethics Committee of Stockholm (numbers 2009/1644/31/3 and 2016/1465-31/4). The patient gave written informed consent in cases where he or she had the capacity to do so. In the majority of cases, the patient lacked capacity, and the patient’s nearest relative gave consent.

## 3. Results

### 3.1. Demographics

At 3 months, 74 patients were able to complete BNIS, and out of these, 41 patients also completed MPAI-4 at the long-term follow-up (Figure 1). Of these 41 patients, 38 had data on educational level, and 33 of them also had TMT data. MPAI-4 was completed by the patients themselves in 22 cases and in 13 cases by rehabilitation personnel. For 6 patients, MPAI-4 was completed both by rehabilitation personnel and the patients themselves. In these cases, the differences between scores were usually small (median = 2 points).

Descriptive statistics of demographic variables, neuropsychological screening scores, MPAI-4, and GOSE for the included patients are presented in Table 1. The high education group was significantly younger than the low education group. No other significant differences were found between educational groups, although according to Cohen’s *d*, there was a large effect size of educational level on BNIS and TMT scores.

### 3.2. Screening Instruments and MPAI-4 in Relation to Demographic Variables

Significant correlations between demographic variables and BNIS score, Cognitive Flexibility_(TMTB-TMTA)_, and MPAI-4 can be seen in Table 2. There were no significant gender differences on any of the measures. In the high education group, seven patients (28%) were above the cut-off value of 47, indicating no cognitive dysfunction, compared to one patient (8%) in the low education group.

### 3.3. Relationships between Neuropsychological Screening and MPAI-4

There were significant correlations between both BNIS and Cognitive Flexibility_(TMTB-TMTA)_ and BNIS and total score on MPAI-4 (Table 3). Dividing MPAI-4 into subscales revealed that it was primarily the participation subscale that was related to BNIS score, while for Cognitive Flexibility_(TMTB-TMTA)_, this correlation was driven by the adjustment and ability subscales.

### 3.4. Regression Analysis

Linear regression was used to assess the independent effect of BNIS and Cognitive Flexibility_(TMTB-TMTA)_ on functional outcome. When adjusting for acute injury severity and educational level (Table 4), linear regression analysis showed that BNIS was significantly and independently related to outcome according to the participation subscale on MPAI-4; Cognitive Flexibility_(TMTB-TMTA)_, however, was not. For the MPAI-4 ability and adjustment subscales, Cognitive Flexibility_(TMTB-TMTA)_ but not BNIS was significantly and independently related to outcome (Table 4). For the adjustment subscale, acute injury severity was also independently and significantly related to outcome in the model both with BNIS (estimate = −0.200, *p* = *0*.020) and with Cognitive Flexibility_(TMTB-TMTA)_ (estimate = −0.2246, *p* = *0*.006). Educational level was not significantly and independently related to outcome in any of the models.

When Cognitive Flexibility_(TMTB-TMTA)_, CRASH, and educational level were included in the model, the MPAI-4 adjustment subscale model was found to be statistically significant (F(3,25) = 5.73, *p* = *0*.004). No other models reached statistical significance.

## 4. Discussion

The present study investigated the relationship between findings from early cognitive screening using BNIS and TMT and long-term outcome assessed with MPAI-4 in sTBI by considering whether this is influenced by cognitive reserve, age, and injury severity. We found that both BNIS and Cognitive Flexibility_(TMTB-TMTA)_ correlated with long-term outcome after sTBI measured with a MPAI-4 total score. When considering the MPAI-4 subscales, we found that BNIS correlated with the participation subscale of MPAI-4, while Cognitive Flexibility_(TMTB-TMTA)_ mainly correlated with the adjustment and ability subscale. Adjusting for cognitive reserve and acute injury severity did not significantly alter these relationships. The results also show, except for the model regarding the adjustment scale containing the variables Cognitive Flexibility_(TMTB-TMTA)_, CRASH, and educational level, that very little of the variance was explained by these variables. This strongly suggests that there are other variables, besides results on cognitive screening, age, and acute injury severity, that influence outcome after sTBI.

Nonetheless, these results suggest that early screening with TMT and BNIS can be used as a tool to help predict later outcomes for patients with sTBI and that these two tests seem to measure different aspects of outcome. Scores on BNIS are related to participation, both independently and when adjusting for confounders. The participation subscale mainly reflects community integration, including a return to work, and our results are in line with previous research emphasizing the link between cognition and return to work [26]. Cognitive Flexibility_(TMTB-TMTA)_ did not correlate significantly with the participation subscale.

The ability subscale in MPAI-4 measures problems with both motor and cognitive abilities and did not correlate with the BNIS score, neither on its own nor when adjusting for controlling variables. The ability scale does, however, correlate with Cognitive Flexibility_(TMTB-TMTA)_. A possible explanation is that Trail Making requires more motor skills compared with BNIS, thereby requiring more of the abilities that the ability scale is measuring. Still, one might have expected a correlation between BNIS and the ability scale since both measure cognitive abilities. The lack of relationship might be due to the fact that rating problems with cognitive abilities are not the same as measuring cognitive function. Prior research has found that subjective measures of cognitive problems have a stronger relationship with a concurrent emotional state than with objective cognitive test measures [27]. An alternative explanation is that individuals with better cognitive function participate more in daily life, according to the correlation between scores on BNIS and the MPAI participation subscale. Greater participation, for instance, more leisure activities, and higher rate of employment likely lead to greater cognitive demands and, therefore, possibly the same amount of experienced cognitive problems in spite of having higher ability.

There was no relationship between BNIS and the MPAI adjustment index, which rates problems with psychological well-being and social interaction. However, acute severity of injury (CRASH) and the score for Cognitive Flexibility_(TMTB-TMTA)_ were both moderately related to outcome on the adjustment scale, acute injury severity slightly stronger than Cognitive Flexibility_(TMTB-TMTA)_. The relationship between the CRASH index and the MPAI adjustment index indicates that CRASH not only can predict mortality but is also related to functional outcome, as have been seen in previous studies [28]. Impaired executive functions have also previously been demonstrated to affect functional outcome negatively, including social functions [14], and difficulties seem to increase with more severe forms of TBI [29]. Executive function is a broad concept that also encompasses aspects such as impulse control, emotion regulation, and motivational drive [12]. In this study, only one aspect of executive function was examined, namely, cognitive flexibility. In order to learn more about the relationship between executive impairments and outcome on MPAI-4 subscales, a more detailed assessment of the executive function would be needed.

We found that as early as 3 months after sTBI, a large effect, according to Cohen’s *d*, could be seen between education groups on both BNIS score and Cognitive Flexibility_(TMTB-TMTA)_. These results were not statistically significant, however, and probably due to a lack of power. This finding is in line with previous research, which has pointed out that the effect of education on cognitive measures is strong even in the early stages after a brain injury, increasing the possibility of misclassifying patients with longer education as having lesser consequences of their brain injury [9,10,30,31]. In our study, 28% of the patients with longer education were classified as having no obvious cognitive impairments from their brain injury 3 months after sTBI, compared with 8% (one patient) in the low education group. In the present study, there was also an age difference between the educational groups, and some of the difference in BNIS scores might be due to the high education group being younger, as there was a correlation between BNIS score and age. Taken together, our findings support the risk of misclassifying patients with longer education as having no cognitive impairments using BNIS, especially if they are also younger.

In the present study, there was no significant difference on outcome according to MPAI-4 based on the level of cognitive reserve, which differs from that of previous studies. However, several previous studies have primarily used cognitive measures as outcome variables, which are more directly influenced by cognitive reserve [32,33]. Nonetheless, other studies have found that cognitive reserve also influences long-term functional outcome after TBI, although these studies have used less detailed outcome assessments [6,34].

### Study Limitations and Strengths

The main limitation of the present study is the small sample size. This limits the conclusions that can be drawn from the data, and the study cannot on its own make definite assumptions on the relationship between neuropsychological screening and long-term functional outcome. Even though all patients were followed by their local outpatient rehabilitation unit, their medical follow-up and rehabilitative efforts varied during the follow-up period. In order to as fully as possible reflect this group of patients, we were restrictive with exclusion criteria; the only one was death or expected death within 3 weeks. This, however, implies that there was less control of medical comorbidities that may have occurred during this time period. The inability to control for further life events, both medical and other, is an inherent flaw in many studies following patients over several years and also greatly restricts the conclusions that can be made from the data.

Initially, over 100 patients were recruited. When considering the incidence of sTBI and the size of the population in Sweden and Iceland, this could in comparative terms be considered a large sample, even though in absolute terms it is small. Given that not all patients could complete a neuropsychological screening and the expected drop-out rate over time, obtaining a larger sample size for a long-term follow-up of sTBI in Sweden is difficult. Drop-out analysis revealed that the only significant difference between included and excluded patients was that included patients had a less severe brain injury according to their GCS score, probably related to the fact that patients able to complete BNIS at 3 months are less severely injured. This highlights another limitation of the study; the results are only generalizable to patients able to complete BNIS at 3 months. The generalizability is also limited to Sweden or countries with a similar social welfare system. An additional weak point is the unusual age difference between the educational groups. However, age was adjusted in the linear regression analysis. A strength of the study is the prospective design; the patients were followed from time of injury until the 5- to 8-year follow-up. We also applied a more nuanced estimate of acute injury severity, using CRASH instead of GCS, thereby better controlling for this variable.

## 5. Conclusions

These findings indicate that for patients able to complete screening with BNIS as well as TMT 3 months after sTBI, these screening instruments are valuable tools that help estimate patients’ long-term outcome after sTBI. Nevertheless, given the small sample size, the results should be interpreted with caution. The instruments measure different aspects of cognition and seem to relate to different aspects of outcome, thereby complementing each other well. Both are relatively easily administered tests that do not require extensive training to use, but consideration should be given to educational level when interpreting neuropsychological test scores. It would be of value to develop education and age-separated norms for both BNIS and TMT. As executive functions include a broad range of functions, it would also be of interest to further explore the impact of various executive impairments on long-term outcome after sTBI.

## Figures and Tables

**Figure 1 jcm-11-02046-f001:**
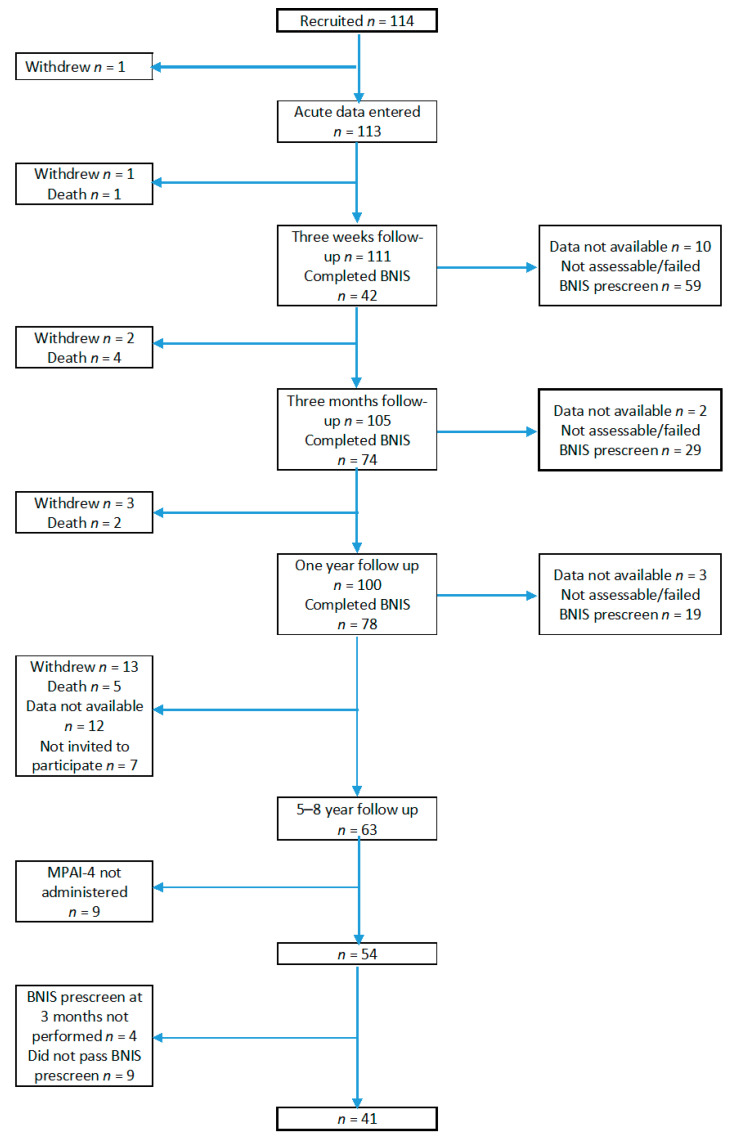
Flowchart of the Probrain long-term follow-up study.

**Table 1 jcm-11-02046-t001:** Descriptive statistics for all participants, separated by educational level.

	Total *n* = 41	Low Education *n* = 13	High Education *n* = 25	Effect Size Cohen’s *d*
Age *	35.6 (13.7)	46.5 (11.2)	35.6 (13.7)	0.85
Gender (M/F)	30/11	11/2	17/8	
CRASH	73 (55.5–83)	76 (49–80)	66 (56–85)	0.006
BNIS total score *	42 (35–46)	39 (34–42)	44 (39–47)	0.66
Cognitive Flexibility _(TMTB-TMTA)_ **	53 (32–82)	68 (32–97)	43 (32–59)	0.85
MPAI-4 total score **	21 (5–36)	19 (5–40)	22 (3–31)	0.20
MPAI-4 ability **	7 (2–16)	4 (2–13)	6 (2–14)	0.06
MPAI-4 adjustment **	8 (1–17)	7 (1–18)	8 (1–15)	0.08
MPAI-4 participation *	5 (0–8)	6 (0–12)	4 (0–7)	0.30
GOSE **	7 (5–8)	6 (5–8)	7 (5–8)	0.04

Note: Values are displayed as median and interquartile range (IQR) for nonparametric data and mean and standard deviation for parametric data (age). TMT missing data are 2 in the low education group and 6 in the high education group. Mann–Whitney was used for examining differences between the groups, except for age, where a Student’s *t*-test was used. * value at 3 months’ follow-up. ** value at 5–8 years’ follow-up.

**Table 2 jcm-11-02046-t002:** Correlation for neuropsychological tests and MPAI-4 with age and injury-related variables.

	BNIS *n* = 41	Cognitive Flexibility _(TMTB-TMTA)_ *n* = 32	MPAI-4 Total Score *n* = 41	MPAI-4 Ability *n* = 41	MPAI-4 Adjustment *n* = 41	MPAI-4 Participation *n* = 41
Age	−0.33 *	0.052	−0.14	−0.23	−0.20	0.13
GCS	0.041	0.097	0.14	−0.19	−0.053	−0.22
CRASH with CT	−0.007	0.11	−0.38 *	−0.36 *	−0.44 **	−0.26

* = *p* < 0.05, ** = *p* < 0.01.

**Table 3 jcm-11-02046-t003:** Correlation between MPAI and neuropsychological screening instruments.

	MPAI-4 Total Score	MPAI-4 Ability	MPAI-4 Adjustment	MPAI-4 Participation
BNIS (*n* = 42)	−0.32 *	−0.28	−0.15	−0.39 *
Cognitive flexibility _(TMTB-TMTA)_ (*n* = 32)	0.36 *	0.36 *	0.36 *	0.34 ^§^

* = *p* < 0.05, ^§^ = *p* = 0.06.

**Table 4 jcm-11-02046-t004:** Linear regression investigating the relationship between neuropsychological screening instruments and MPAI, adjusting for acute injury severity (including age) and educational level.

		Unadjusted		Adjusted		R-Square
		Est.	95% CI	Est.	95% CI	Adjusted Model
MPAI-4 ability	BNIS	−0.32	−0.72–0.077	−00.26	−0.73–0.20	0.12
	Cognitive Flexibility _(TMTB-TMTA)_	0.060 **	0.016–0.10	0.059 *	0.0026–0.11	0.25
MPAI-4 adjustment	BNIS	−0.10	−0.50–0.30	−0.18	−0.66–0.30	0.18
	Cognitive Flexibility _(TMTB-TMTA)_	0.068 **	0.020–0.12	0.076 **	0.021–0.13	0.41
MPAI-4 participation	BNIS	−0.32 *	−0.59–0.057	−0.41 *	−0.74–−0.074	0.21
	Cognitive Flexibility _(TMTB-TMTA)_	0.033 *	0.00034–0.067	0.029	−0.012–0.070	0.26

* = *p* < 0.05, *** = p <* 0.01. Separate linear regression models for BNIS and cognitive flexibility for each MPAI-4 outcome scale. In the adjusted models, CRASH and educational level are used as adjusting variables.

## Data Availability

Data available on request due to restrictions (e.g., privacy or ethical). The data presented in this study are available on request from the corresponding author. The data are not publicly available due to the fact that the ethical board requires the data to be kept confidential in order to protect the privacy of the patients.

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
