# Peer review of "Cognitive Reserve, Early Cognitive Screening, and Relationship to Long-Term Outcome after Severe Traumatic Brain Injury"

_jcm, 2022, doi:10.3390/jcm11072046_

Round 1

Reviewer 1 Report

This topic is not so common, but interesting. Please look at these points to improve:

  • Lines 47-48 ("objective was to investigate the relationship between early global cognitive functioning and cognitive flexibility") and 74-78 ("primary aim was to investigate the relationship between findings from early cognitive screening") sounds similar. Are both sentences necessary? 
  • This study has a very long follow-up (5-8 years), this should be highlighted in the introduction section when the aim is exposed.
  • Lines 38-39: "The most severe form of TBI (sTBI) is characterized by great variance in outcome, from death to favorable outcome." Some traumatic lesion are at high risk of death. Look at these two related refs:  --  Posttraumatic synchronous double acute epidural hematomas: Two craniotomies, single skin incision. Surg Neurol Int. 2020 Dec 11;11:435. doi: 10.25259/SNI_697_2020. --   Traumatic acute convexity interdural hematoma: a case report and literature review. Br J Neurosurg. 2020 May 2:1-3. doi: 10.1080/02688697.2020.1749985. 
  • Are your results similar or differ from other recent studies? Please discuss about it in the discussion seciton. Look at refs. : -- The Association Between Brain Reserve, Cognitive Reserve, and Neuropsychological and Functional Outcomes in Males With Chronic Moderate-to-Severe Traumatic Brain Injury. Am J Speech Lang Pathol. 2021 Apr 16;30(2S):883-893.  -- The Influence of Cognitive Reserve on Recovery from Traumatic Brain Injury. Arch Clin Neuropsychol. 2019 Mar 1;34(2):206-213. 
  • Lines 247-249 : "Cognitive Flexibility(TMTB-TMTA) did not correlate significantly with the participation subscale. This might be due to a smaller sample-size, as the strength of the correlation was very close to the pre-set significance level." What do authors mean in these sentences?
  • Lines 262-263: "This results in them experiencing a similar level of difficulty even though they have a higher ability" How do authors explain these results?
  • Lines 265-267: "However, both acute severity of injury (CRASH) and the score for Cognitive Flexibility(TMTB-TMTA) were related to outcome on the adjustment scale" Which of the two events has the greatest impact on the outcome? Please improve.

Reviewer 2 Report

Study of 41 patients with severe TBI who completed the BNIS at 3 months (33 of which completed the TMT at 3 months), and the MPAI-4 at 5-8 years. The authors found that BNIS correlated with the MPAI participation subscale, and TMT correlated with the adjustment and ability subscales. From a language standpoint the study is well written, and it seeks to address an important problem between subacute and long-term outcomes.

The study unfortunately suffers from very small sample size which limits the degree of multivariable control. The authors should describe the rationale behind using the MPAI vs. a battery of measures at 5-8 years, which is a long time after index TBI. Why was the MPAI chosen as a solitary measure, and what is the justification to use 3 month outcomes to predict outcomes >5 years from index injury? Is the MPAI the measure of choice for 5-8 year outcomes after TBI? The correlation with TMT is from an even smaller sample size.With an N of 30-40, the Limitations section of this paper must be dramatically improved given the significant number of limitations. Other factors such as medical comorbidities, rehabilitative efforts, medical follow-up, and socioeconomic support are all factors that influence long term outcome. This study has significant limitations and will require a much larger sample size to come up with conclusive data. As such, it should be reformatted into a brief report or letter to the editor.

Round 2

Reviewer 1 Report

Good.

Reviewer 2 Report

Limitations augmented